# SORNet: Spatial Object-Centric Representations for Sequential Manipulation

**Wentao Yuan**
University of Washington

**Chris Paxton**
NVIDIA

**Karthik Desingh**
University of Washington

**Dieter Fox**
University of Washington, NVIDIA

**https://wentaoyuan.github.io/sornet**

**Abstract:** Sequential manipulation tasks require a robot to perceive the state of an environment and plan a sequence of actions leading to a desired goal state, where the ability to reason about spatial relations among object entities from raw sensor inputs is crucial. Prior works relying on explicit state estimation or end-to-end learning struggle with novel objects or new tasks. In this work, we propose **SORNet** (**S**patial **O**bject-Centric **R**epresentation **Net**work), which extracts object-centric representations from RGB images conditioned on canonical views of the objects of interest. We show that the object embeddings learned by SORNet generalize *zero-shot* to *unseen* object entities on three spatial reasoning tasks: spatial relation classification, skill precondition classification and relative direction regression, significantly outperforming baselines. Further, we present real-world robotic experiments demonstrating the usage of the learned object embeddings in task planning for sequential manipulation.

**Keywords:** Object-centric Representation, Spatial Reasoning, Manipulation

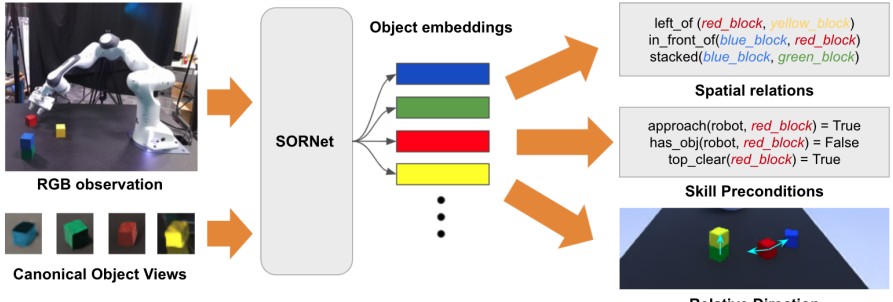

Figure 1: We propose **SORNet** (**S**patial **O**bject-Centric **R**epresentation **Net**work) that learns object embeddings useful for spatial reasoning tasks such as predicting spatial relations, classifying skill preconditions, and regressing relative direction between objects.

## 1 Introduction

A crucial question for complex multi-step robotic tasks is how to represent relations between entities in the world, particularly as they pertain to preconditions for various skills the robot might employ. In goal-directed sequential manipulation tasks with long-horizon planning, it is common to use a state estimator followed by a task and motion planner or other model-based system [1, 2, 3, 4, 5, 6]. A variety of powerful approaches exist for explicitly estimating the state of objects in the world, e.g. [7, 8, 9, 10]. However, it is challenging to generalize these approaches to an arbitrary collection of objects. In addition, the objects are often in contact in manipulation scenarios, where works explicitly addressing the problem of generalizing to unseen objects [11, 12] still struggle.

5th Conference on Robot Learning (CoRL 2021), London, UK.

Fortunately, knowing exact poses of objects may not be necessary for manipulation. End-to-end methods [13, 14, 15] leverage that fact and build networks that generates actions directly from sensor inputs without explicitly representing objects. Nevertheless, these networks are very specific to the tasks they are trained on. For example, it is non-trivial to use a network trained on stacking blocks to unstack blocks.

In this work, we take an important step towards a manipulation framework that generalizes zero-shot to novel tasks with unseen objects. Specifically, we propose a neural network that extracts object-centric embeddings from raw RGB images conditioned on object queries, which we call **SORNet**, or **S**patial **O**bject-Centric **R**epresentation **Net**work. The design of SORNet allows it to generalize to novel objects without retraining or finetuning. The object-centric embeddings produced by **SORNet** can be combined with readout networks to inform a task and motion planner with implicit object states relevant to goal-directed sequential manipulation tasks, e.g. logical preconditions for primitive skills or continuous 3D directions from the end effector to objects in the scene.

To summarize, our contribution are: (1) a method for extracting object-centric embeddings from RGB images that generalizes zero-shot to different number and type of objects; (2) a framework for learning object embeddings that capture continuous spatial relations with only logical supervision; (3) a dataset containing sequences of RGB observations labeled with spatial predicates during various tabletop rearrangement manipulation tasks.

We empirically evaluate the object-centric embeddings produced by **SORNet** on three different downstream tasks: 1) classification of logical predicates for action preconditions; 2) visual question and answering for spatial relations; 3) prediction of relative 3D direction between entities in manipulation scenes. In all three tasks, the models are tested on held-out objects that did not appear in training data. In all tasks, **SORNet** obtains significant improvements over the baseline methods. Finally, we evaluated SORNet with real-world robot experiments to showcase the transfer of the learned object-centric embeddings to real-world observations.

## 2   Related Work

**Sequential Manipulation** In goal-directed sequential manipulation tasks with long-horizon planning, a class of work uses a pipeline approach requiring a state estimator followed by a task and motion planner [1, 2, 3, 4, 5, 6], but the state estimator with an explicit state representation is often restricted to an environment or a set of objects. Another class of work employs neural networks to learn motor controls directly from raw sensor data, such as RGB images and joint encoder readings [14, 16, 15, 17, 18, 19], but they are often not transferable to new tasks, especially tasks involving long-horizon planning. Our work combines a network that estimates implicit states with a symbolic planner to obtain the best of both worlds: generalization to new objects and to new tasks.

**Learning Spatial Relations** Learning spatial relations between object entities have been studied in the field of 3D vision and robotics. Methods such as [20, 21, 22] predict discrete or continuous pairwise object relations from 3D inputs such as point clouds or voxels, assuming complete observation of the scene and segmented objects with identities. In contrast, our approach does not make any assumptions regarding the observability of the objects and does not require pre-processing of the sensor data. The learning framework by Kase et al. [23] is most related to our approach, which takes a sequence of sensor observations and classifies a set of pre-defined relational predicates which is then used by a symbolic planner to produce a suitable operator. Compared to our approach, theirs is limited by the number of objects in the scene and a fixed set of spatial predicates.

**Visual Reasoning** Recently, several advancements have been made on visual reasoning benchmarks [24, 25, 26] using transformer networks [27]. Toward solving spatio-temporal reasoning task from CLEVRER [25] and CATER [26], Ding et al. [28] proposed an object-based attention mechanism and Zhou et al. [29] proposed a multi-hop transformer model. Both works assume a segmentation model to produce object segments and performs language grounding to the segments to perform reasoning. Our SORNet architecture is simpler and can solve spatial-reasoning tasks for unseen object instances without requiring a segmentation or object detection module. Furthermore, our work focuses on a relatively complex manipulation task domain involving manipulator in the observations. Although our current work focuses on predicting spatial relations from a single RGB frame, the object-centric embeddings could potentially be used for solving temporal-reasoning tasks.

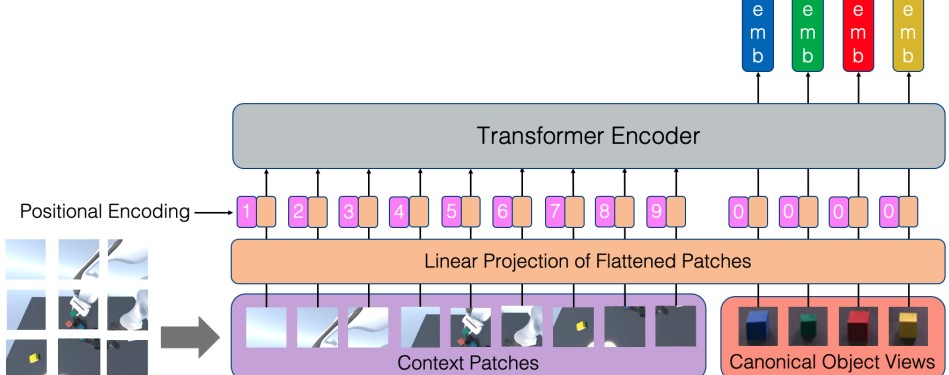

Figure 2: **SORNet** architecture. Input to the network is an RGB image and canonical views of the objects of interest. The RGB image is broken into context patches which have the same size as the canonical views. These patches are flattened and added with positional encoding and passed through a multi-layer multi-head transformer [30]. The embeddings corresponding to the canonical views are used for the downstream tasks.

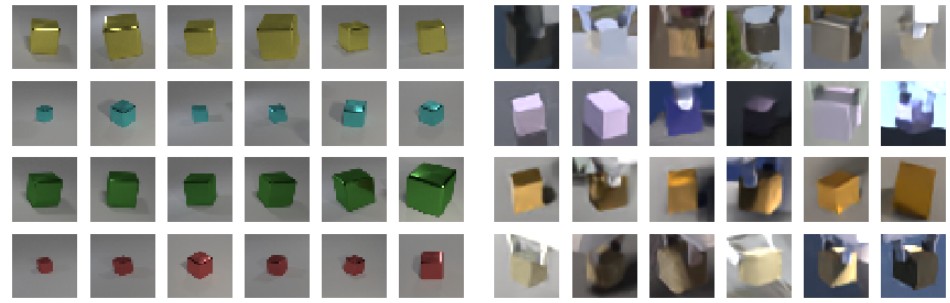

Figure 3: Examples of canonical object views used for training CLEVR (left) and Leonardo(right) dataset. Lighting, texture and object poses vary across different views, and sometimes there is occlusion by the robot.

## 3 Methods

### 3.1 SORNet: Spatial Object-Centric Representation Network

Our object embedding network (**SORNet**) (Fig. 2) takes an RGB image and an arbitrary number of canonical object views and outputs an embedding vector corresponding to each input object patch. The architecture of the network is based on the Visual Transformer (ViT) [30]. The input image is broken into a list of fixed-sized patches, which we call *context patches*. The context patches are concatenated with the canonical object views to form a patch sequence. Each patch is first flattened and then linearly projected into a token vector, then positional embedding is added to the sequence of tokens. Following [30], we use a set of learnable vectors with the same dimension as the token vectors as positional embeddings. The positional-embedded tokens are then passed through a transformer encoder, which includes multiple layers of multi-head self-attention. The transformer encoder outputs a sequence of embedding vectors. We discard the embedding for context patches and keep those for the canonical object views.

We apply the same positional embedding to the canonical object views to make the output embeddings permutation equivariant. We also mask out the attention among canonical object views and the attention from context patches to canonical object views to ensure the model uses information from the context patches to make predictions. In this way, we can pass in an arbitrary number of canonical object views in arbitrary order without changing model parameters during inference.

Intuitively, the canonical object views can be viewed as queries where the context patches serve as keys to extract the spatial relations' values. Note that the canonical object views are *not* crops from the input image, but arbitrary views of the objects that may not match the objects' appearance in the scene. Our model learns to identify objects even under drastic change in lighting, pose and occlusion. Fig. 3 shows some examples of canonical object views used in our experiments.

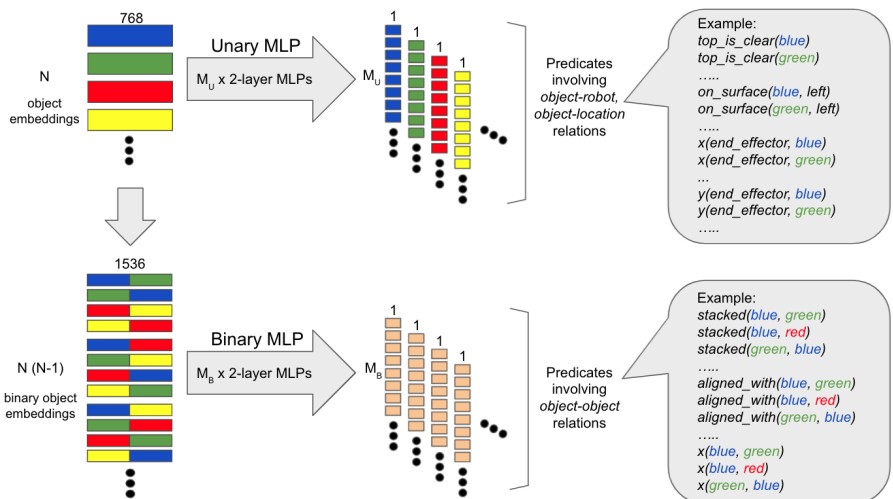

Figure 4: Architecture of the readout networks, which uses the object embeddings from **SORNet** to predict spatial relations, such as logical statements that can serve as skill preconditions or continuous 3D directions. The readout network is flexible to accommodate any number of input object embeddings without changing its parameters. $M_U$ and $M_B$ denote the number of unary and binary relation types respectively.

## 3.2 Readout Networks

The readout networks (Fig. 4) are responsible for predicting a list of relations using object embeddings. The relations can be logical statements, e.g., whether the blue block is stacked onto the green block, or continuous quantities, e.g. which direction should the end effector move to reach the red block. The readout networks consist of a collection of 2-layer MLPs, one for each type of relations. Here we focus on unary and binary relations. Unary relations involve a single object or an object and the environment, which could be the robot or a region on the table. Binary relations involve two objects and, optionally, the environment. In principle, our framework is extensible to relations involving more than two objects, but we leave that for future work.

The readout network for unary relations takes the list of object embeddings and outputs relations pertaining to the object that the embedding is conditioned on. Taking the top_is_clear classifier for an example, if the input embedding is conditioned on the blue block, the network will output whether there is any object on top of the blue block. If the input embedding is conditioned on the red block, the network will output whether there is any object on top of the red block.

The readout network for binary relations takes a list of binary object embeddings created by concatenating pairs of object embeddings and outputs relations corresponding to a pair of objects, e.g., whether the blue block is on top of the red block. Thus, with $N$ object embeddings, there will be $N(N-1)$ binary object embeddings and $N(N-1)$ output relations.

Parameters of the readout network are independent of the number of objects. The number of output relations dynamically changes with the number of input object embeddings. For example, when are 7 unary relations and 2 binary relations, with 4 objects, the network generates $7 \times 4 + 2 \times 4 \times (4-1) = 52$ outputs; with 5 objects, the network generates $7 \times 5 + 2 \times 5 \times (5-1) = 75$ outputs. In this way, our overall model generalizes zero-shot to scenes with an arbitrary number of objects.

## 4 The Leonardo Dataset

To test the generalization of our model to new objects and tasks, we created a simulated tabletop environment named Leonardo, where a Franka Panda robot manipulates a set of randomly colored blocks. The robot is given a goal formulated as a list of predicates to be satisfied, and then use a simple task planner [5] to find a sequence of actions to achieve that goal and a sample-based motion planner [31] to generate trajectories and choose grasps. As we know the ground truth poses of the blocks in the simulator, we can compute ground-truth logical predicates at every step of the planning process. We used NVISII [32] to render the RGB observations. Domain randomization including random lighting, background and perturbations to the camera position is applied while rendering.

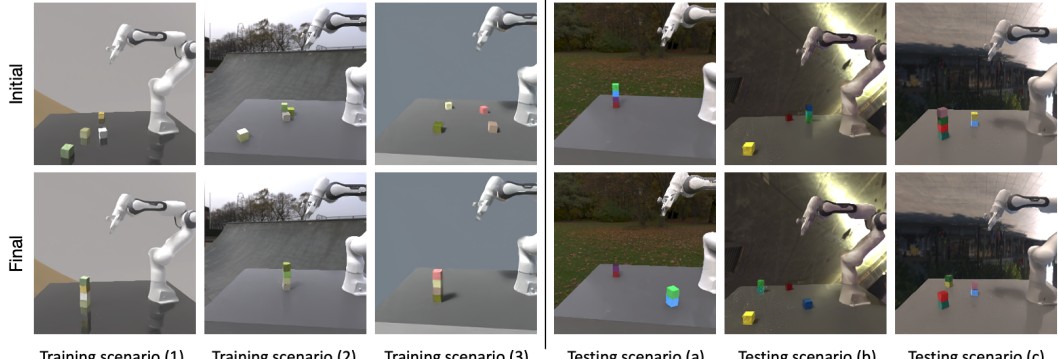

Figure 5: Sample scenes from training and testing scenarios in Leonardo dataset. Top row shows the initial configuration of a task scenario and the bottom row shows the goal condition. The training scenarios contains 4 blocks with random colors with a single tower stack as a goal condition. The testing scenarios contain 4-6 blocks with specific colors with various goal conditions involving multi-tower stacking scenarios.

The training data contains 133796 sequences of a single task - stacking 4 blocks in a tower. The block colors are randomly chosen from 405 xkcd colors[1]. The testing data contains 9526 sequences with 4-6 blocks, chosen from colors that are not included in the training data: *red, green, blue, yellow, aqua, pink, purple*. The testing tasks consists of 7 tasks different from training. Fig. 5 shows some examples. Please see the supplement for full description of the testing tasks.

## 5 Results

### 5.1 CLEVR-CoGenT

We first evaluate our approach on a variant of the CLEVR dataset [24], a well established benchmark for visual reasoning. CLEVR contains rendered RGB images with at most 10 objects per image. There are 96 different objects in total (2 sizes, 8 colors, 2 materials, 3 shapes). Each image is labeled with 4 types of spatial relations (right, front, left, behind) for each pair of objects.

Specifically, we use the CoGenT version of the dataset, which stands for Compositional Generalization Test, where the data is generated in two different conditions. In condition A, cubes are gray, blue, brown, or yellow and cylinders are red, green, purple, or cyan. Condition B is the opposite: cubes are red, green, purple, or cyan and cylinders are gray, blue, brown, or yellow. Spheres can be any color in both conditions. The models are trained on condition A and evaluated on condition B. The training set (trainA) contains 70K images and the evaluation set (valB) contains 15K images. Several prior works [33, 34] show significant generalization gap on CLEVR-CoGenT caused by the visual model learning strong spurious biases between shape and color.

We generate a question for each spatial relation in the image, e.g. "*Is the large red rubber cube in front of the small blue metal sphere?*" We filter out any query that is ambiguous, e.g. if there were two large red cubes, one in front and one behind the small blue sphere. This results in around 2 million questions for both valA and valB sets. We compare against MDETR [34], which reports state-of-the-art zero-shot result on CLEVR-CoGenT, i.e. there is no fine-tuning on any example from condition B. The results are summarized in Table 1. Our model performs drastically better on classifying spatial relations of unseen objects and shows a much smaller generalization gap between valA and valB sets.

Unlike MDETR which takes text queries, our model takes visual queries in the form of canonical object views (i.e. 2 canonical views for the objects mentioned in the question). To eliminate the influence of those factors, we report the performance of the MDETR model trained on the full CLEVR dataset, denoted as MDETR-oracle. We can see that although SORNet is only trained on condition A, it is able to achieve similar performance to MDETR-oracle. The zero-shot generalization ability of our model can potentially be combined with other reasoning pipelines to improve generalization performance on other types of queries as well.

---

[1] https://xkcd.com/color/rgb/

|  | MDETR [34] | MDETR-oracle [34] | **SORNet(ours)** |
|---|---|---|---|
| ValA Accuracy | 84.950 | 97.944 | **99.006** |
| ValB Accuracy | 59.627 | 98.052 | **98.222** |

Table 1: Zero-shot relation classification accuracy on CLEVR-CoGenT [24]. The MDETR-oracle model has seen all the objects during training, where as MDETR and SORNet have only see objects in condition A. SORNet takes canonical views as queries whereas MDETR takes text queries.

## 5.2 Predicate Classification

Next, we evaluate the task of predicate classification on the Leonardo dataset. We compare against 3 baselines that do not use object conditioning. The first two baselines use a ResNet18 and a ViT-B/32 respectively to directly predict 52 predicates. The last baseline uses the same architecture as ours, but the embedding tokens come from 4 fixed class embedding vectors. We report 3 metrics: accuracy, F-1 score and all-match accuracy across all predicates and among predicates within a category. The accuracy and F-1 score are computed per predicate and averaged. For all-match accuracy, the predicates are considered as a single vector, which most faithfully represents how the predicates are treated by a planner. If any predicate is classified incorrectly, the prediction for the entire category is considered as wrong. The models are tested on images where the objects are completely unseen during training.

The results are summarized in Table 2. Models labeled with M-View are trained on 3 different views of the scene as data augmentation. During testing, we aggregate the predictions from 3 views by adding the logits. The M-View models are better at handling occlusions by leveraging multi-view information. The gripper state is concatenated to the object embedding in the **SORNet** variant denoted by (G) in Table 2, which is better at classifying predicates correlated to the opening/closing of the gripper, e.g. `has_obj`. We can see that the non-object-conditioned baselines fail drastically when applied to unseen objects zero-shot. Even after fine-tuning on 100 examples, they still significantly underperform zero-shot SORNet. This demonstrates the generalizability of our model obtained via conditioning on canonical object views.

Further, we tested our model on scenes with 5 to 6 objects, while it has only been trained on 4-object scenes. We first treat the additional objects as distractors, which shows that it is not necessary to acquire canonical views of every object in the scene if they are not of interest. We then treat all objects as objects of interest. In this case, the number of binary predicates increases quadratically with the number of objects in the scene, so the all-match accuracy naturally drops even if the average accuracy and F-1 score remain the same. Note that none of the baselines can even be applied to these scenes with more objects without introducing additional model parameters and retraining the model.

Finally, we run our best performing model on 30 real-world images of a robot performing various manipulation tasks. The quantitative results are in Table 2 and Fig. 6 shows two qualitative examples of the predicates predicted by our model. Our model transfers to real-world without losing much accuracy. It does make some mistakes when encountered with novel scenarios never seen during training, such as one block stacked in between two blocks (right plot in Fig. 6).

## 5.3 Skill Executability

We further evaluate the predicate prediction in the context of task planning. Each frame in the Leonardo dataset is labeled with a primitive skill that will be executed, e.g. grasp the red block. Each skill has a list of preconditions that needs to be satisfied before it can be executed, which can be formulated as a vector of predicate values. In other words, if all of the predicates in the preconditions are classified correctly, we can determine the executability of that skill correctly. In Table 3, we report the accuracy for classifying the executability of skills in the Leonardo test set using the predicate predictions.

This evaluation puts more emphasis on the predicates relevant to the manipulation of objects, e.g. `approach`, `aligned`, which are rarely true in the training data. Our model is able to identify these predicates correctly whereas the baselines fail completely on skills relevant to object manipulation, i.e. `grasp`, `align`, `place` and `lift`.

| | | Accuracy | | | | | | |
|---|---|---|---|---|---|---|---|---|
| Method | # pred | all | on_surface | has_obj | top_clear | stacked | aligned | approach |
| Majority | 52 | 88.2 | 79.3 | 88.4 | 75.1 | 91.7 | 99.2 | 92.9 |
| ResNet18 M-Head 0-shot | 52 | 88.5 | 80.0 | 88.4 | 76.2 | 91.7 | 99.2 | 92.9 |
| ViT-B/32 M-View 0-shot | 52 | 88.5 | 80.2 | 88.4 | 76.2 | 91.7 | 99.2 | 92.9 |
| ViT-B/32 M-Head M-View 0-shot | 52 | 88.5 | 80.2 | 88.4 | 76.0 | 91.7 | 99.2 | 92.9 |
| ResNet18 M-Head 100-shot | 52 | 88.5 | 80.0 | 88.4 | 76.4 | 91.7 | 99.1 | 92.9 |
| ViT-B/32 M-View 100-shot | 52 | 88.6 | 79.9 | 88.3 | 78.7 | 91.7 | 99.1 | 92.8 |
| ViT-B/32 M-Head M-View 100-shot | 52 | 92.8 | 89.9 | 90.4 | 87.8 | 92.4 | 99.2 | 93.5 |
| SORNet 0-shot | 52 | 97.6 | 96.9 | 95.3 | 96.6 | 98.7 | 99.2 | 96.1 |
| SORNet M-View 0-shot | 52 | **98.9** | **99.0** | 95.9 | **99.2** | **99.6** | **99.4** | **97.3** |
| SORNet M-View (G) 0-shot | 52 | **98.9** | 98.9 | **98.8** | 98.5 | 99.5 | 99.3 | 96.6 |
| SORNet M-View 1 distractor 0-shot | 52 | 98.3 | 98.6 | 96.6 | 95.2 | 98.6 | 99.5 | 97.5 |
| SORNet M-View 2 distractors 0-shot | 52 | 97.6 | 98.2 | 96.7 | 89.9 | 97.7 | 99.5 | 97.3 |
| SORNet M-View (G) 5 obj 0-shot | 70 | 98.5 | 98.5 | 99.4 | 95.8 | 98.2 | 99.6 | 97.4 |
| SORNet M-View (G) 6 obj 0-shot | 102 | 98.0 | 98.3 | 99.6 | 93.9 | 96.8 | 99.7 | 97.7 |
| SORNet M-View (G) real-world 0-shot | 52 | 96.3 | 96.4 | 96.7 | 93.3 | 97.1 | 96.7 | 95.6 |

| | | All-match Accuracy | | | | | | |
|---|---|---|---|---|---|---|---|---|
| Method | # pred | all | on_surface | has_obj | top_clear | stacked | aligned | approach |
| ResNet18 M-Head 0-shot | 52 | 0.0 | 0.3 | 53.5 | 30.4 | 30.1 | 89.8 | 71.6 |
| ViT-B/32 M-View 0-shot | 52 | 0.0 | 0.4 | 53.5 | 30.3 | 30.1 | 89.8 | 71.6 |
| ViT-B/32 M-Head M-View 0-shot | 52 | 0.0 | 0.3 | 53.5 | 30.3 | 30.1 | 89.8 | 71.6 |
| ResNet18 M-Head 100-shot | 52 | 0.0 | 0.3 | 53.5 | 30.9 | 30.1 | 89.8 | 71.6 |
| ViT-B/32 M-View 100-shot | 52 | 0.2 | 3.3 | 55.0 | 40.5 | 31.0 | 89.8 | 72.5 |
| ViT-B/32 M-Head M-View 100-shot | 52 | 4.1 | 21.7 | 62.4 | 63.8 | 40.0 | 89.8 | 75.0 |
| SORNet 0-shot | 52 | 43.8 | 66.4 | 81.4 | 87.8 | 87.9 | 90.5 | 86.1 |
| SORNet M-View 0-shot | 52 | 60.2 | **86.3** | 83.5 | **96.8** | **95.8** | **92.4** | **89.5** |
| SORNet M-View (G) 0-shot | 52 | **63.6** | 84.5 | **95.8** | 94.0 | 94.3 | 92.0 | 86.6 |
| SORNet M-View 1 distractor 0-shot | 52 | 50.1 | 80.9 | 86.3 | 82.7 | 85.9 | 94.4 | 91.0 |
| SORNet M-View 2 distractors 0-shot | 52 | 39.2 | 76.3 | 87.2 | 68.2 | 78.3 | 95.0 | 90.4 |
| SORNet M-View (G) 5 obj 0-shot | 70 | 45.5 | 74.8 | 97.6 | 81.1 | 72.6 | 92.0 | 87.5 |
| SORNet M-View (G) 6 obj 0-shot | 102 | 29.7 | 68.9 | 97.7 | 70.2 | 52.4 | 92.0 | 87.5 |
| SORNet M-View (G) real-world 0-shot | 52 | 26.7 | 63.3 | 93.3 | 80.0 | 76.7 | 90.0 | 80.0 |

| | | F-1 Score | | | | | | |
|---|---|---|---|---|---|---|---|---|
| Method | # pred | all | on_surface | has_obj | top_clear | stacked | aligned | approach |
| ResNet18 M-Head 0-shot | 52 | 9.7 | 23.6 | 0.0 | 31.5 | 0.0 | 0.0 | 0.0 |
| ViT-B/32 M-View 0-shot | 52 | 11.9 | 37.4 | 0.0 | 28.9 | 0.0 | 0.0 | 0.1 |
| ViT-B/32 M-Head M-View 0-shot | 52 | 12.2 | 32.5 | 0.0 | 27.7 | 0.0 | 0.0 | 0.0 |
| ResNet18 M-Head 100-shot | 52 | 0.0 | 21.9 | 0.0 | 32.6 | 0.0 | 0.0 | 0.0 |
| ViT-B/32 M-View 100-shot | 52 | 0.0 | 37.7 | 6.3 | 46.5 | 0.0 | 0.0 | 7.3 |
| ViT-B/32 M-Head M-View 100-shot | 52 | 0.0 | 70.5 | 31.0 | 73.2 | 27.2 | 0.0 | 23.2 |
| SORNet 0-shot | 52 | 83.2 | 92.2 | 79.7 | 93.0 | 91.2 | 63.8 | 74.9 |
| SORNet M-View 0-shot | 52 | 88.9 | **97.5** | 82.0 | **98.4** | **97.3** | **70.5** | **81.7** |
| SORNet M-View (G) 0-shot | 52 | **89.5** | 97.1 | **94.7** | 96.8 | 96.4 | 69.9 | 76.7 |
| SORNet M-View 1 distractor 0-shot | 52 | 85.1 | 96.3 | 81.8 | 90.1 | 88.8 | 67.5 | 80.1 |
| SORNet M-View 2 distractors 0-shot | 52 | 80.2 | 95.2 | 80.2 | 79.4 | 81.4 | 60.7 | 75.3 |
| SORNet M-View (G) 5 obj 0-shot | 70 | 85.3 | 96.0 | 96.7 | 91.3 | 83.6 | 69.8 | 78.1 |
| SORNet M-View (G) 6 obj 0-shot | 102 | 79.9 | 95.5 | 97.0 | 87.5 | 69.2 | 70.0 | 77.9 |
| SORNet M-View (G) real-world 0-shot | 52 | 76.5 | 90.7 | 85.7 | 80.3 | 69.1 | 33.3 | 68.9 |

Table 2: Zero-shot predicate classification results on the Leonardo test data where the objects are held-out from training. SORNet significantly outperforms baselines that are not object-conditioned, and is able to generalize to scenes with a different number of objects without retraining or finetuning.

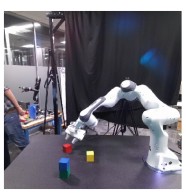
on_surface(green_block, left)
on_surface(red_block, right)
on_surface(red_block, far)
on_surface(yellow_block, center)
top_is_clear(red_block)
top_is_clear(blue_block)
top_is_clear(yellow_block)
in_approach_region(robot, red_block)
stacked(green_block, blue_block)

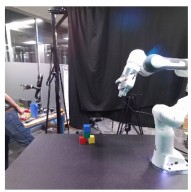
on_surface(red_block, right)
on_surface(yellow_block, right)
top_is_clear(red_block)
top_is_clear(blue_block)
stacked(red_block, green_block)
stacked(green_block, blue_block)
stacked(yellow_block, green_block)

Figure 6: Visualization of predicted predicates in real-world scenes. Black means true positive, blue means false positive and red means false negative. True negatives are not shown due to limited space.

| | Accuracy | | | | | | |
| Method | all | approach | grasp | align | place | lift | go_home |
|---|---|---|---|---|---|---|---|
| ResNet18 M-View | 27.0 | 88.7 | 0.0 | 0.0 | 0.0 | 0.0 | 100.0 |
| ViT-B/32 M-View | 27.4 | 90.6 | 0.0 | 0.0 | 0.0 | 0.0 | 100.0 |
| ViT-B/32 M-Head M-View | 27.3 | 90.4 | 0.0 | 0.0 | 0.0 | 0.0 | 100.0 |
| SORNet | 63.7 | 64.3 | 72.4 | 67.4 | 99.4 | 53.0 | 97.6 |
| SORNet M-View | 63.6 | 65.1 | 69.1 | 99.8 | 52.9 | 69.7 | 99.5 |
| SORNet M-View (G) | 76.3 | 98.7 | 68.1 | 99.9 | 99.9 | 69.7 | 100.0 |

Table 3: Skill executability accuracy on the Leonardo test data. SORNet is the only model that is able to correctly determine the executability of skills relevant to the handling of objects.

| Method | ResNet18 | ResNet18 (MV) | ResNet18 (P) | CLIP-ViT | CLIP-ViT (P) | SORNet (P) | SORNet (P MV) |
|---|---|---|---|---|---|---|---|
| Obj-Obj | 0.4308 | 0.6068 | 0.9876 | 0.9875 | 0.6145 | 0.1679 | **0.1458** |
| EE-Obj | 0.3251 | 0.3464 | 0.5929 | 0.6544 | 0.4960 | 0.1962 | **0.1777** |

Table 4: Euclidean error on regression of continuous 3D unit vector between entities in the scene. The regressors are trained on 1000 examples with unseen objects. Methods labeled with P are pretrained on the Leonardo dataset. Methods labeled with MV use 3 views.

## 5.4 Open Loop Planning

In this demo, we incorporate **SORNet** as a part of an open-loop planning pipeline in a real-world manipulation scenario. Specifically, given an initial frame, we use the predicates predicted by SORNet M-View (G) to populate a state vector. A task and motion planner takes the state vector and desired goal (formulated as a list of predicate values to be satisfied), and outputs a sequence of primitive skills. The robot then executes this sequence of skills in an open loop fashion. This demonstrate that how SORNet can be applied to sequential manipulation of unseen object in a zero-shot fashion, i.e. without any fine-tuning on the test objects. Please refer to our supplementary video for the demo.

## 5.5 Relative Direction Prediction

Although the training objective of SORNet is purely logical, with large-scale pre-training, the object embeddings learned by SORNet contains continuous spatial information. We demonstrate this fact by using SORNet embeddings to predict the relative 3D direction between entities. Specifically, we trained a regressor (same architecture as the classifier in Sec. 3.2) on top of frozen SORNet embeddings to predict the continuous direction between two objects (Obj-Obj) or the direction the end effector should move to reach a certain object (EE-Obj). The regressor is trained using L2 loss on a 1000 examples with unseen objects, and tested on 3000 examples with the same objects.

The results are summarized in Table 4. SORNet is able to outperform models trained from scratch with pose supervision (ResNet18) as well as models initialized with weights from large-scale language-image pretraining [35] (CLIP-ViT). This demonstrates that our representation learning technique is more suited to manipulation scenarios where more precise spatial information is crucial. In our supplementary video, we showed that by predicting the direction for movement in an online fashion, our model can be used for visual servoing to guide the robot to reach a target object.

## 6 Conclusion

We proposed **SORNet** (**S**patial **O**bject-Centric **R**epresentation **Net**work) that learns object-centric representations from RGB images. We show that the object embeddings produced by **SORNet** capture spatial relations which can be used in a downstream tasks such as spatial relation classification, skill precondition classification and relative direction regression. Our method works on scenes with an arbitrary number of unseen objects in a zero-shot fashion. With real-world robot experiments, we demonstrate how **SORNet** can be used in manipulation of novel objects.

**Acknowledgments**

This work was supported in part by the National Science Foundation under Contract NSF-NRI-2024057 and in part by Honda Research Institute as part of the Curious Minded Machine initiative.

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
