# OpenReview forum: "SORNet: Spatial Object-Centric Representations for Sequential Manipulation"
_robot-learning.org/CoRL/2021/Conference — CoRL2021 Oral_

### Official Review · Reviewer_eyTp · 2021-07-23

**Originality:** Good
**Technical Quality:** Good
**Clarity Of Presentation:** Good
**Impact:** 3

**Recommendation:**

Strong Accept: I recommend accepting the paper and will argue for my recommendation even if other reviewers hold a different opinion.

**Summary:**

The paper presents Spatial Object-Centric Representation Network (SORNet), an approach that takes an RBG image and several canonical object views, and outputs object-centric embeddings which are then used in 3 spatial reasoning tasks, including: spatial relationship classification, skill precondition classification and relative direction regression. In addition to this, real-world robot manipulation experiments are shown to demonstrate the ability to use these representations for task planning.


**Issues:**

**Questions**
- After you get the plan, how do you locate the pose of the blocks in the real world? This detail seems missing in the real-world section.

**Reviewer Expertise:**

Good: General knowledge of the area

**Strengths And Weaknesses:**

**Strengths**

- The system offers a flexible way to accommodate a number of objects in the scene.
- The paper is clear and presents thorough results.
- Convincing real world results.

**Weaknesses**

- This list of contributions seem superfluous. To recap, the states contributions are (1) architecture, (2) dataset, (3) experiments, (4) real robot demo. It's not clear to me how the architecture is a contribution, as it is essentially just a Visual Transformer (ViT). Moreover, experiments to back up your method should not be counted as a contribution; these are required in any paper to support a presented method. I would argue that there are only 2 real contributions here: the method and the dataset.
- Related work could use a bit more attention; currently only has a small paragraph each on spatial and visual reasoning. What about considering other sequential manipulation papers that don't fall within these areas, and explain why perhaps Spatial Object-Centric Representations are better suited for the style pf problem?
- Relative Direction Prediction experiment need to be motivated more; it's not clear why this is  useful thing to learn and doesn’t seem to fit with the rest of the paper. Perhaps this can be expanded upon. However, the video does briefly motivate with a visual servoing section.

**Minor**
- Missing domain rand citations (particularly in the manipulation domain); more details on the domain randomisation? Amount of noise, etc
- Typo "… direction between in the 3D space …"

**Summary Of Recommendation:**

The paper has limited technical novelty however, the proposed system instead gives strong experimental results.

---

> ### Author Response · Authors · 2021-08-24
> **Response to Review by Reviewer eyTp**
>
> Thanks for the great suggestions. We have corrected the typos and updated the paper as you suggested. Please find our responses to your questions below.
> - **The list of contributions seem superfluous.**
>   - We’ve revised the list of contributions as follows: (1) a method for extracting object-centric embeddings from RGB images that generalizes zero-shot to different number and type of objects; (2) a framework for learning object embeddings that capture continuous spatial relationships with only logical supervision; (3) a dataset containing sequences of RGB observations labeled with spatial predicates during various tabletop rearrangement manipulation tasks.
> - **What about considering other sequential manipulation papers that don't fall within these areas, and explain why perhaps Spatial Object-Centric Representations are better suited for the style of problem?**
>   - We have cited a few sequential manipulation papers [1-6]. The purpose of SORNet is exactly to replace the explicit object state in these works with implicit neural object representations. We have added a discussion on this in the related work section.
> - **The relative Direction Prediction experiment needs to be motivated more.**
>   - The relative direction prediction experiment demonstrates SORNet’s capability in predicting continuous quantities useful for controlling the robot (e.g. visual servoing), while pre-trained only on logical predicates. This is very important for future work as the current work focuses on predicting high-level actions and does not generate low-level motion commands. We have added more details for the visual servoing experiment in the updated paper to better motivate this task.
> - **After you get the plan, how do you locate the pose of the blocks in the real world?**
>   - We used PoseCNN [7] to get object poses, from which we generate low-level motion for primitive actions like grasping. The purpose of the real-world demo is to show the high-level reasoning component from our model transfers well from simulation to the real robot. As mentioned above, in future work we could utilize SORNet’s capability in predicting continuous quantities such as the relative direction between objects to handle the low-level component as well.
>
> [7] Xiang, Yu, et al. "Posecnn: A convolutional neural network for 6d object pose estimation in cluttered scenes." arXiv preprint arXiv:1711.00199 (2017).

---

> > ### Comment · Reviewer_eyTp · 2021-08-30
> > **Reviewer Response**
> >
> > Thank you for your response and answering my questions/concerns. I am happy to increase my rating.

---

### Official Review · Reviewer_zJy5 · 2021-07-23

**Originality:** Good
**Technical Quality:** Good
**Clarity Of Presentation:** Good
**Impact:** 4

**Recommendation:**

Strong Accept: I recommend accepting the paper and will argue for my recommendation even if other reviewers hold a different opinion.

**Summary:**

This paper addresses the problem of classifying objects' spatial relationships from vision. To do this, it proposes a novel architecture which takes in the scene image along with close ups of each object and embeds each object view and each image patch into an embedding space. A transformer jointly processes all of these embeddings to output one embedding per object, which can then be passed to downstream tasks, such as classifying spatial relations. The results show a high level of performance on a block rearrangement task and on the clevr dataset.

**Issues:**

The paper does not compare to any existing object-attention based methods, nor does it show any results with realistic objects (vs solid colored, convex, geometrical shapes). Although the paper writes that [24] "assume a segmentation model", my understanding is that [24] uses an unsupervised learning method to learn the segmentations. Other than this, the model looks very similar to SORNet, so some further explanation of sets SORNet apart would help explain the novelty.


**Reviewer Expertise:**

Very good: Comprehensive knowledge of the area

**Strengths And Weaknesses:**

Strengths:
- The proposed method enables learning object agnostic relationships by conditioning on the specific objects at inference time.
- The real world robot results show that this model can transfer well form sim to real and is robust enough for planning through stacking tasks.
- The clevr results show the model generalizes to novel object-attribute compositions.
- While this model will not work for all manipulation tasks, it looks like this could be used to quickly put together controllers for new tasks which could be useful for robotics practitioners.

Limitations:
- As proposed, the method does not address the case of objects appearing different from different angles, or the case of needing to take the object's pose into account when manipulating it (e.g. a trapezoid or a long prism).
- The results as shown require a high supervision burden of labeling object relationships explicitly. Many prior approaches in this area (e.g. relation networks, [25]) instead aim to learn these relationships implicitly from sparser supervision.

Questions:
- If the objects look different at different orientations, a single canonical view could be insufficient for finding the object in the scene. I wonder if simply averaging the embeddings from multiple views would be enough to address this limitation. This would be an interesting experiment to improve the paper.



**Summary Of Recommendation:**

I like the method and overall results. The model is novel but not groundbreakingly so, and has only been shown to work in highly supervised settings. However, the object generalization and real world results are promising for future work.

+++
The update from the authors has addressed my comments and I can strongly recommend acceptance.

---

> ### Author Response · Authors · 2021-08-24
> **Response to Review by Reviewer zJy5**
>
> Thank you for your valuable comments. Please let us know if our responses below address your concerns.
> - **As proposed, the method does not address the case of objects appearing different from different angles, or the case of needing to take the object's pose into account when manipulating it (e.g. a trapezoid or a long prism).**
>   - We would like to emphasize that the appearance of objects in canonical views may not match with the appearance of objects in the scene image. Object poses in both canonical views and the scene images are randomized during training. Thus, SORNet does learn to match the objects appearing from different viewing angles. We believe that SORNet is able to handle shapes like trapezoid and long prism with a single canonical view if it is trained on a large set of objects. We agree that training the network with more complex shapes is an important next step.
> - **If the objects look different at different orientations, a single canonical view could be insufficient for finding the object in the scene.**
>   - We believe that by training on a sufficiently large set of objects with randomized views, SORNet is capable of identifying objects looking different in different poses with a single canonical view. This would be as true as it is with language; both language and a single canonical view under-specify the object’s appearance in different ways. In the future, we could also try to feed SORNet with multiple canonical views of the same object and fuse the predictions.
> - **I wonder if simply averaging the embeddings from multiple views would be enough to address this limitation.**
>   - We did use multiple views in our experiments (the M-view models in Table 2, see supplement for a visualization of the different views). Averaging the embeddings is a valid option, but we chose to perform majority voting on the predicate predictions from different views, because our models are trained to predict the predicates using embedding from one random view, instead of averaged embedding from multiple views.
> - **The results as shown require a high supervision burden of labeling object relationships explicitly.**
>   - The current supervision is indeed quite dense, but these supervision signals are cheap in simulation, and we were able to show sim-to-real transfer results. We are also looking at ways to use sparser supervision, such as only supervising with the pre/post conditions of each action, but this will be addressed in future work.
> - **The paper does not compare to any existing object-attention based methods, nor does it show any results with realistic objects (vs solid colored, convex, geometrical shapes). Although the paper writes that [24] "assume a segmentation model", my understanding is that [24] uses an unsupervised learning method to learn the segmentations. Other than this, the model looks very similar to SORNet, so some further explanation of sets SORNet apart would help explain the novelty.**
>   - The focuses of Aloe [24] and SORNet are very different. Aloe consumes object embeddings produced by MONet [1] and uses them to reason about things like object dynamics from a video. SORNet produces object embeddings. In fact, the object embedding from SORNet can be supplied as input to Aloe, replacing the MONet embeddings. We would like to compare to MONet under the setup of Aloe in the future, but their implementation is not publicly available yet.
>   - Compared to SORNet, the disadvantage of unsupervised segmentation methods like MONet is that MONet needs to segment out every object in the scene. This is simple for CLEVR-style scenes but very challenging for manipulation scenarios with a lot of clutter and an articulated robot arm. In fact, we have never seen works like Aloe and MONet being tested on real-world images. In contrast, SORNet can take in objects of interest as queries and ignore background and clutter without requiring good-quality segmentation. which makes SORNet work on scenes with more occlusions and much easier to deploy to the real world.
>
> [1] Burgess, Christopher P., et al. "Monet: Unsupervised scene decomposition and representation." arXiv preprint arXiv:1901.11390 (2019).
>
> [24] Ding, David, et al. "Object-based attention for spatio-temporal reasoning: Outperforming neuro-symbolic models with flexible distributed architectures." arXiv preprint arXiv:2012.08508(2020).

---

> > ### Comment · Reviewer_zJy5 · 2021-08-27
> > **Follow-up**
> >
> > Thank you for your reply, which has answered most of my questions.
> >
> > I would like to clarify two of my points however.
> >
> > “As proposed, the method does not address the case of objects appearing different from different angles”
> > Your answer, that the model will work for objects looking different from different angles because it is trained to do, is correct for the set of training objects. However, SORNet claims that “the object embeddings learned by SORNet generalize zero-shot to unseen object entities on three spatial reasoning tasks”. Consider an object that is red from one side and green from another. If SORNet was trained with this object, it would learn to recognize that the two viewpoints are the same object. However, if this object was introduced only at test time (zero shot to an unseen object), there is no reason for the model to know that the green object it was given in the canonical viewpoint corresponds to the red object in the scene, especially if there are other objects of the same shape also in the scene. This limitation isn’t very important (it would be difficult to have any model generalize to this situation), but it is indeed something the model would struggle with.
> >
> > “ the case of needing to take the object's pose into account when manipulating it (e.g. a trapezoid or a long prism).”
> > What I am referring to here is not the perception aspect, but rather the behavior aspect of the experiments. My understanding is that the robot manipulation experiments are all with cubes. The motions used to manipulate the objects are generated from a planner. Planning motions to grasp cubes is relatively easy, aiming for the center of mass will almost always work. Planning to grasp non-cubes generally requires a 3D mesh of the object, which would not be available for “unseen object entities”.

---

> > > ### Author Response · Authors · 2021-08-30
> > > **Response to Follow-up**
> > >
> > > **“As proposed, the method does not address the case of objects appearing different from different angles” Your answer, that the model will work for objects looking different from different angles because it is trained to do, is correct for the set of training objects. However, SORNet claims that “the object embeddings learned by SORNet generalize zero-shot to unseen object entities on three spatial reasoning tasks”. Consider an object that is red from one side and green from another. If SORNet was trained with this object, it would learn to recognize that the two viewpoints are the same object. However, if this object was introduced only at test time (zero shot to an unseen object), there is no reason for the model to know that the green object it was given in the canonical viewpoint corresponds to the red object in the scene, especially if there are other objects of the same shape also in the scene. This limitation isn’t very important (it would be difficult to have any model generalize to this situation), but it is indeed something the model would struggle with.**
> > >   - Very good point. You are correct that with a single canonical view and a single view of the scene, the model won’t be able to handle objects with completely different appearance from different views if it is not trained with such objects. In the current work, what we have done to mitigate this issue is to provide multiple views of the scene, hoping that most of the views can reveal that the object is part red and part green, so the model can match it with the canonical view. In future work, we could extend SORNet to take multiple canonical views of objects in order to identify the objects more robustly.
> > > **“the case of needing to take the object's pose into account when manipulating it (e.g. a trapezoid or a long prism).” What I am referring to here is not the perception aspect, but rather the behavior aspect of the experiments. My understanding is that the robot manipulation experiments are all with cubes. The motions used to manipulate the objects are generated from a planner. Planning motions to grasp cubes is relatively easy, aiming for the center of mass will almost always work. Planning to grasp non-cubes generally requires a 3D mesh of the object, which would not be available for “unseen object entities”.**
> > >   - This is a great observation. Manipulating objects with more complex geometry requires accurate information on the objects’ geometry and orientation in addition to their location, but we don’t think 3D meshes are absolutely necessary. There are many works that grasp objects without 3D meshes, e.g. [2]. We would also like to highlight that SORNet does understand object geometry and orientation, which are required to complete the stacking task. To precisely stack two blocks, the robot needs to align the blocks precisely before placing a block on top of another, otherwise the tower will be unstable or misaligned. Our model supplies information about object geometry and orientation to the planner via the “aligned_with” predicates, which classifies whether the object in hand is aligned with the chosen grasp. We plan to further explore the capability of SORNet on manipulation of more complex objects in future work.
> > >
> > > [2] Mousavian, Arsalan, Clemens Eppner, and Dieter Fox. "6-dof graspnet: Variational grasp generation for object manipulation." Proceedings of the IEEE/CVF International Conference on Computer Vision. 2019.

---

### Official Review · Reviewer_nXEV · 2021-07-25

**Originality:** Excellent
**Technical Quality:** Good
**Clarity Of Presentation:** Excellent
**Impact:** 4

**Recommendation:**

Strong Accept: I recommend accepting the paper and will argue for my recommendation even if other reviewers hold a different opinion.

**Summary:**

This paper introduces a network architecture based on vision-transformers to produce object-centric latent embeddings, conditioned on canonical views of the objects in question. The learned embeddings are shown to be useful to estimate logical predicates that can be used for question answering and symbolic planning. Particularly interesting is their generalization to novel objects, settings, and from simulation to a real robot setting.

**Issues:**

## CLEVR experiments

L159: For the 2 million questions, are only the validation parts of condition A and B counted? How many questions are used for training?

How are the canonical object views generated here? As mentioned in the paper, providing these, simplifies the task compared to the baseline as the network does not need to infer visual appearance from text. This discrepancy and the fact that the evaluation does not use the questions from the dataset should be clarified in Table 1. Are there baselines that also take in canonical views and therefore would allow a better direct comparison?

Also, are the questions turned into predicates for the proposed method but asked as text to MDETR? Could this cause issues in the comparison?

Table 1: Why does MDETR-oracle achieve a higher ValA accuracy that MDETR? Shouldn’t training on both conditions only help in condition B? Condition A is already used in the non-oracle version, right? Or is this version also trained on validation data in addition to training data? Either way, it would be good to clarify this in Table 1. Also, would it make sense to add a SORNet-oracle variant to this comparison? Why or why not?

L167-169: Based on the results, I agree that the bottleneck is generalizability, but I do not think that the data proves that “generalizability of the visual model” is the bottleneck. It could also be generalizability between the concrete scenarios and questions, e.g. between training and validation splits.

## Predicate Classification

How do the baseline methods classify predicates for unseen objects (L184) if they are not conditioned on these objects?

Table 2: The fact that the baselines do not outperform simple majority prediction indicates that there might be an issue with training the baselines on the imbalance dataset. Did the authors attempt to address label imbalance, e.g. by adapting the sampling between positives and negatives or by weighing the loss terms appropriately?

## Relative Direction Prediction

I am not sure how the relative direction prediction task is formulated here, especially for the baselines. Are the objects fixed and the models always predict the direction between all pairs of objects?

Table 5b shows that the baselines overfit here, which is not surprising given the small dataset size used in this experiment. However, there are a number of ways to reduce overfitting, such as early stopping, regularization losses, dropout etc. Did the authors apply any of these techniques to reduce overfitting in the baselines?


**Reviewer Expertise:**

Good: General knowledge of the area

**Strengths And Weaknesses:**

The paper presents a very interesting idea that is at the heart of where current approaches to robot learning struggle: generalization to unseen objects and settings. The proposed method is novel but not overly complex (which is great). The method is tested creatively in a variety of scenarios including on a real robot. The paper is also very well organized and written. I was not able to find some details of the experiments (see below) but I am sure that this can be easily fixed.

The part that I am most worried about are the different baselines. As the paper uses novel tasks or task variations that others have not published on, it is important to make sure that the baselines are the right ones and that they have been properly tweaked and optimized. Based on the text, I am not fully convinced that this is the case here (see below).

**Summary Of Recommendation:**

This is an overall very interesting paper with novel ideas that could have a significant impact on the field. Right now, there is some information missing from the experimental section and the baselines have to be improved. If these issues are resolved, I would be happy to update my rating accordingly.

EDIT: The issues were addressed by the rebuttal.

---

> ### Author Response · Authors · 2021-08-24
> **Response to Review by Reviewer nXEV**
>
> Thank you for your detailed feedback! We hope we have answered the questions you raised to your satisfaction below.
> - **For the 2 million questions, are only the validation parts of condition A and B counted? How many questions are used for training?**
>   - Yes, the 2M questions are for validation. 978M questions are used for training.
> - **How are the canonical object views generated here?**
>   - The canonical object views are cropped from the images. Each object has 2-14 canonical views with different poses and lighting. During training, we randomly sample canonical views, but during inference, a single canonical view for each object is used for the entire evaluation.
>   - We would like to note that objects in the canonical views do not look the same as the objects in the scene image. The objects may be viewed from a different angle, lighted differently or occluded. Our network needs to identify the objects under drastic appearance variations.
> - **As mentioned in the paper, providing canonical views simplifies the task compared to the baseline as the network does not need to infer visual appearance from text. This discrepancy and the fact that the evaluation does not use the questions from the dataset should be clarified in Table 1.**
>   - We have made a note of the difference in Table 1. However, we would like to note that although our model does not infer visual appearance from text, it does need to match the appearance between objects in the canonical view and objects in the image, which can be significantly different under different lighting, texture and object pose and occlusion. We would not say that canonical views are always simpler than textual descriptions. In future work, we plan to incorporate language in addition to using canonical images.
>   - We would also like to note that objects in the real world can not always be easily described, but canonical views can be provided easily by e.g. taking a picture with your phone visually showing the object of interest. This makes the visual queries we use more applicable to real-world manipulation scenarios than text queries.
> - **Are there baselines that also take in canonical views and therefore would allow a better direct comparison?**
>   - We were not aware of any other baselines that are able to achieve the level of zero-shot adaptation like SORNet using canonical views. We welcome suggestions for more baselines we could compare to, and will incorporate the comparisons if time permits.
> - **Are the questions turned into predicates for the proposed method but asked as text to MDETR? Could this cause issues in the comparison?**
>   - The texts we supplied to MDETR are a templated language that is easy to parse, e.g. “Is [the large brown metal cube] to the left of [the small red rubber sphere]?”, where the texts inside the brackets are replaceable. Since we use the same canonical view for each object during evaluation, the canonical views can be thought of as a replacement for phrases like “the large brown metal cube”. Just like MDETR, SORNet also needs to identify the query objects in the image and reason about their relationships with other objects. Hence, this setup should not cause issues in the comparisons we discussed in the paper.
> - **Why does MDETR-oracle achieve a higher ValA accuracy that MDETR?**
>   - This is an interesting observation. The training set for MDETR is smaller than that for MDETR-oracle and systematically biased. This may have affected MDETR’s ability to learn informative embeddings. We will check with the authors of MDETR to see if they have more insights.
> - **Would it make sense to add a SORNet-oracle variant to this comparison?**
>   - After fixing an issue in the data loader after the original submission, SORNet is able to achieve similar performance as MDETR oracle (99.006 on valA, 98.222 on valB), even if it was only trained on condition A. Training on both conditions A and B will likely give similar results. Thus, we don’t think adding a SORNet-oracle variant is necessary.
> - **Based on the results, I agree that the bottleneck is generalizability, but I do not think that the data proves that “generalizability of the visual model” is the bottleneck. It could also be generalizability between the concrete scenarios and questions, e.g. between training and validation splits.**
>   - Good point. To be more specific, what we mean is the generalizability of the visual model to novel objects and different scene configurations, which bring systematic shifts to the distribution. It is different from the kind of generalizability in e.g. ImageNet classification, where the task is the same between training and testing.

---

> > ### Author Response · Authors · 2021-08-24
> > **Response to Review by Reviewer nXEV**
> >
> > - **How do the baseline methods classify predicates for unseen objects if they are not conditioned on these objects?**
> >   - The baselines predict predicates for a fixed set of objects, ignoring their identities. For example, if the training object is pink while the testing object is red, they output predicates about the red object as if it is pink in the 0-shot case. We agree that the baseline methods are not designed for 0-shot generalization, so we added 100-shot results for the baselines to the updated paper, where the baselines are fine-tuned on 100 sequences with the testing objects.The fine-tuned baselines outperform majority prediction but underperform SORNet.
> > - **The fact that the baselines do not outperform simple majority prediction indicates that there might be an issue with training the baselines on the imbalance dataset. Did the authors attempt to address label imbalance, e.g. by adapting the sampling between positives and negatives or by weighing the loss terms appropriately?**
> >   - Yes, we applied weighted sampling to the training data. In particular, we assign a weight to each image that is inversely proportional to the rarity of the most rare predicate value. For example, if a predicate is true 1 out of 50 times, images with that predicate will have weight 50. We will add more details in the appendix. The fact that the baselines do not outperform majority prediction is because they are not designed for zero-shot generalization. As mentioned above, 100-shot fine-tuning results for the baselines are added to the updated paper.
> > - **I am not sure how the relative direction prediction task is formulated here, especially for the baselines. Are the objects fixed and the models always predict the direction between all pairs of objects?**
> >   - For the baselines, yes, the objects are fixed and the model predicts the relative directions between all pairs of objects in the image. However, SORNet is more flexible and predicts relative directions between the query objects (i.e. objects in the canonical views). For the purpose of comparison, we just query all the objects in the image.
> > - **There are a number of ways to reduce overfitting, such as early stopping, regularization losses, dropout etc. Did the authors apply any of these techniques to reduce overfitting in the baselines?**
> >   - We have tried applying early stopping and L2 weight decay, but they have minimal effect on the baselines’ performance. One thing that did improve the ViT-B/32 baseline is to use pre-trained CLIP [1] weights instead of starting from scratch. We have included updated results in the revised paper.
> >
> > [1] Radford, Alec, et al. "Learning transferable visual models from natural language supervision." arXiv preprint arXiv:2103.00020 (2021).

---

> > > ### Comment · Reviewer_nXEV · 2021-08-27
> > > **Re: Response**
> > >
> > > Thank you for the clarifications. Since this addresses my main concerns, I'll update my rating accordingly.

---

### Meta-Review · Area_Chair_ihQc · 2021-08-13

**Recommendation:** Accept (Oral)
**Confidence:** 5

**Metareview:**

Post-rebuttal, there is clear and consistent agreement from all three reviewers that this new method is useful and novel, and potentially important. This is a well-executed paper built around a cool idea that follows through to real-robot experiments.

---

> ### Author Response · Authors · 2021-08-24
> **Response to Meta Review by Area Chair ihQc**
>
> We would like to thank the area chair and the reviewers for their time. We find your feedback very helpful and have updated the paper with more details about experiments, baselines and related work (updated parts are highlighted in red). We respond to the specific questions from each reviewer in the official comments below.

---

### Decision · Program_Chairs · 2021-09-13

**Decision:**

Accept (Oral)

**Comment:**

Post-rebuttal, there is clear and consistent agreement from all three reviewers that this new method is useful and novel, and potentially important. This is a well-executed paper built around a cool idea that follows through to real-robot experiments.